# Probiotic Milk Enriched with Protein Isolates: Physicochemical, Organoleptic, and Microbiological Properties

**DOI:** 10.3390/foods13193160

**Published:** 2024-10-03

**Authors:** Małgorzata Pawlos, Katarzyna Szajnar, Magdalena Kowalczyk, Agata Znamirowska-Piotrowska

**Affiliations:** Department of Dairy Technology, Institute of Food Technology and Nutrition, College of Natural Sciences, University of Rzeszow, Cwiklinskiej 2D, 35-601 Rzeszow, Poland; kszajnar@ur.edu.pl (K.S.); mkowalczyk@ur.edu.pl (M.K.)

**Keywords:** fermented milk, whey protein isolate, soy protein isolate, pea protein isolate, probiotic, *Lactobacillus johnsonii*, *Lacticaseibacillus casei*, microbiological analysis, color, syneresis, organoleptic evaluation

## Abstract

Incorporating plant protein isolates into milk can enhance probiotic culture growth by providing essential nutrients and altering the physicochemical properties of fermented milk. This study investigated the effects of adding 1.5% or 3.0% soy, pea, and whey protein isolates on the growth of *Lacticaseibacillus casei* and *Lactobacillus johnsonii* monocultures, as well as the physicochemical (acidity, syneresis, color) and organoleptic properties of fermented milk during 21 days of refrigerated storage. The results showed that 1.5% SPI and WPI did not significantly alter milk acidity compared to controls. Still, pH increased with 1.5% and 3.0% PPI. Storage time significantly affected pH in *L. casei* fermented milk. The initial addition of WPI at 1.5% and 3.0% reduced syneresis in *L. casei* fermented milk compared to other samples. Color components were significantly influenced by isolates. Initial *L. casei* cell counts were lower with SPI (LCS1.5 and LCS3) and 1.5% PPI (LCP1.5) compared to controls. Increasing isolate concentration from 1.5% to 3% enhanced *L. johnsonii* growth in WPI-milk but reduced *L. casei* in LCW3 compared to LCW1.5. Only increased pea protein concentration significantly increased *L. casei* growth. Probiotic populations generally were reduced during extended storage. Moreover, isolates impacted milk organoleptic evaluation. This research demonstrates the potential of protein isolates in creating health-promoting and diverse fermented products and offers insights into their interaction with probiotic cultures to advance functional food technologies.

## 1. Introduction

The popularity of plant-based diets and alternative dairy products has significantly grown in recent years, more than doubling global sales between 2009 and 2015, reaching 21 billion USD [1]. Plant-based proteins serve as a nutritious and economically viable source of protein in developing countries, where the consumption of animal protein is limited and costly. For instance, in developed countries, legumes are often regarded as alternative protein sources, particularly critical vegan food ingredients [2]. Additionally, it has been demonstrated that bacteria such as *Lactobacillus*, *Lactococcus*, and *Bifidobacterium* spp. express peptidases and proteolytic enzymes, which play a role in proteolysis within various food matrices. The proteolytic properties of lactic acid bacteria (LAB) have long been utilized in the production of fermented foods. Fermentation of plant matrices with LAB can enhance the digestibility and bioavailability of plant proteins [3,4,5,6]. Therefore, the use of plant protein isolates in the production of fermented milk could enrich the nutritional value of the final product.

Furthermore, the production of fermented milk involves standardizing the fat content and normalizing the non-fat dry matter concentration. Non-fat dry matter, comprising the milk component and non-dairy additives, determines the final product’s texture, contributing to its firmness. In relation to fermented milk, food law (Codex Alimentarius) defines a minimum protein content (not less than 2.7%) but does not specify a maximum limit. Examples of concentrated fermented milk include Stragisto, Labneh, Ymer, and Ylette [7,8]. The normalization of non-fat dry matter can be achieved by thickening milk using an evaporator, membrane techniques, or by the addition of milk protein preparations such as milk powders, milk or whey protein concentrates and isolates, caseinates, proteinates, coprecipitates, and others [9].

Due to their high protein content, pea protein isolate (PPI), soy protein isolate (SPI), and whey protein isolate (WPI) are commonly used in food processing to enhance the quality of food products. Therefore, in this study, we chose to use these isolates as an additional source of protein. The applied plant proteins may improve the structure and consistency of fermented milk and affect syneresis. Additionally, the use of plant protein isolates may enhance the allergenic profile, particularly in the case of pea protein, which is considered an emerging alternative to soy protein, as it is non-genetically modified and has lower allergenicity than soy. The development of technology for producing more nutritious fermented pea-based protein products is highly desirable by both industry and consumers [4].

PPI has a very low allergen content, which is particularly important for individuals with various food intolerances. It contains no lactose, gluten, and nearly no carbohydrates [4,10]. Peas contain 65 to 80% globulin, 10 to 20% albumin, and a high lysine content [11]. Pea protein exhibits antioxidant, antihypertensive, anti-inflammatory, cholesterol-lowering, and bacterial-modulating properties [12,13]. Additionally, it has the ability to foam, emulsify, retain oil, retain water, and is soluble, which influences the texture of food products. 

In contrast, SPI contains at least 90% protein on a dry matter basis. Soy protein provides a good supply of essential amino acids compared to other plant-based protein sources and is characterized by a high lysine content. Soy protein also has emulsifying properties and can modulate the texture of food products [14].

In this study, WPI was also used, primarily for comparative purposes. WPI is the purest form of whey protein, containing 90% or more protein with minimal lactose (<1%) and nearly no fat. WPI has excellent emulsifying properties, which can reduce interfacial tension and tightly coat fat globules, forming a more stable oil–water interface [15]. 

Several studies have confirmed that the fermentation of plant matrices with LAB can increase the release of compounds from the matrix that is available for absorption [3,4,5,6]. Research has shown that the probiotic *Bacillus coagulans* GBI-30, 6086 enhances protein hydrolysis and improves the digestion of soy, pea, and rice proteins in vitro under simulated digestive conditions [16,17].

The protein isolates used in our study can serve as a source of nutrients, including nitrogen and peptides, to support the growth of probiotic bacteria. For probiotic cultures to thrive in the development of fermented products, such as fermented milk, food products containing bacterial cultures are considered probiotic only if they contain a minimum concentration of live bacteria of 6 log CFU g^−1^ [18], which can be achieved by consuming 100 g or 100 mL of food daily [19,20]. Therefore, in the present study, we aimed to verify whether the use of different protein isolates would stimulate the growth of live cells of probiotic monocultures, such as *Lacticaseibacillus casei* and *Lactobacillus johnsonii*. The results of our study may provide new and valuable insights into the specific interactions between different protein isolates and the growth of probiotics.

## 2. Materials and Methods

### 2.1. Materials

The fermented milk was prepared using 2% fat Łaciate milk (microfiltered and pasteurized at 74 °C, 15 s; SM Mlekpol, Grajewo, Poland). 100% SPI, PPI, and WPI “Biały Puch” were purchased from F.H.U. “KDJ” s.c. (Tarnów, Poland). The starter culture of probiotic bacteria *Lactobacillus johnsonii* LJ Delvo^®^Pro was purchased from DSM (Delft, The Netherlands), and *Lacticaseibacillus casei* 431 was provided by Chr. Hansen (Hoersholm, Denmark). MRS agar and peptone water were supplied by Biocorp (Warszawa, Poland). Sodium hydroxide and phenolphthalein were produced by Chempur (Piekary Śląskie, Poland). All reagents used were of analytical grade.

### 2.2. Production of Fermented Milk

The milk was divided into experimental groups, with protein isolates added according to Figure 1. Subsequently, the milk samples containing the isolates, as well as the control milk, were subjected to homogenization (60 °C, 20 MPa) and re-pasteurized at 85 °C for 10 min. Next, the milk samples were cooled to 37 ± 1 °C and inoculated with a single starter culture of *Lactocaseibacillus casei* 431 (CLC—control milk, LCS1.5—milk with the addition of 1.5% SPI, LCS3—milk with the addition of 3% SPI, LCP1.5—milk with the addition of 1.5% PPI, LCP3—milk with the addition of 3% PPI, LCW1.5—milk with the addition of 1.5% WPI, LCW3—milk with the addition of 3% WPI) or *Lactobacillus johnsonii* (CLJ—control milk, LJS1.5—milk with the addition of 1.5% SPI, LJS3—milk with the addition of 3% SPI, LJP1.5—milk with the addition of 1.5% PPI, LJP3—milk with the addition of 3% PPI, LJW1.5—milk with the addition of 1.5% WPI, LJW3—milk with the addition of 3% WPI). Each batch of milk was inoculated with a pre-activated starter culture, activated at 40 °C for 5 h, and then added to the milk at a concentration of 5% (*w*/*w*), following the method described by Szajnar et al. [20]. The inoculated milk was stirred, poured into 100 mL plastic containers, fermented at 37 °C for 12 h, and then cooled to 5 °C (Cooled Incubator ILW 115, POL-EKO Aparatura, Wodzisław Śląski, Poland). The experiment was repeated three times, and the fermented milk was evaluated after the first and twenty-first days of refrigerated storage at 5 °C.

### 2.3. Acidity 

The pH measurement was conducted with a pH-meter (FiveEasy Mettler Toledo, Greifensee, Switzerland) equipped with electrodes InLab^®^Solids Pro-ISM (Mettler Toledo, Greifensee, Switzerland). The milk’s total acidity (TA, g of lactic acid L^−1^) was determined according to the method of Jemaa et al. [21].

### 2.4. Syneresis

Syneresis was measured by assessing the quantity of whey released in relation to the initial weight. A 10 g sample of fermented milk was transferred into a 50 mL plastic tube and subjected to centrifugation using the Refrigerated Centrifuge LMC-4200R (Biosan SIA, Riga, Latvia) at 3160 × g for 10 min, 5 °C [22].

### 2.5. Color

The color analysis of fermented milk was performed using a colorimeter (the Precision Colorimeter, Model NR 145, Shenzhen, China) using the instrumental method based on the CIELab system. The lightness was measured using the parameter L* (0—black, 100—white). Chromaticity was determined using the a* value (−a*—indicating green hues, +a*—indicating red hues) and the b* value (−b*—representing blue hues, + b*—representing yellow hues). Additionally, color saturation and purity were expressed by the C* value, while hue was assessed using the h^0^ parameter. Prior to the measurements, the colorimeter was calibrated against a white reference standard [23].

### 2.6. Microbiological Analysis

For each sample, 10 g were diluted in 90 mL of sterile peptone water solution (0.1%), and serial dilutions ranging from 1 log CFU g^−1^ to 8 log CFU g^−1^ were prepared. Inoculation was carried out through the plate-deep method using MRS agar, followed by anaerobic incubation in a vacuum desiccator at 37 °C for 72 h using the GENbox anaer (Biomerieux, Warszawa, Poland) [24]. Inoculation was carried out using the plate-deep method with MRS agar, followed by anaerobic incubation in a vacuum desiccator placed in an incubator (Cooled Incubator ILW 115, POL-EKO Aparatura, Wodzisław Śląski, Poland) at 37 °C for 72 h, using the GENbox anaer system (Biomerieux, Warsaw, Poland) [24]. The probiotic colonies were counted using a colony counter, TYPE J-3 (Chemland, Stargard, Poland), and the results were expressed as log CFU g^−1^.

### 2.7. Organoleptic Evaluation

The organoleptic evaluation was conducted by a trained panel (17 women and 13 men, aged 23–55). The encoded samples of fermented milk (labeled with random three-digit codes) were assessed on a 9-point rating scale with edge markings. The left end represented the least intense and least characteristic feature, and the right end indicated the most intense and most characteristic feature. The panelists evaluated the following attributes: consistency, milky-creamy taste, sour taste, sweet taste, off-taste, fermentation odor, sour odor, and off-odor [25,26]. Definitions of attributes in the descriptive organoleptic analysis of fermented milk [27]:Milky-creamy taste: the taste stimulated by milk powder.Sour taste: the taste stimulated by lactic acid.Sweet taste: the taste stimulated by sucrose.Off-taste: an unidentified taste that is not characteristic.Fermentation odor: the intensity of odor associated with sour milk and fresh fermented dairy products.Sour odor: the odor stimulated by acids.Off-odor: unidentified odor that is not characteristic.

### 2.8. Statistical Analysis

Based on the acquired data, statistical analysis was conducted using Statistica 13.1 (StatSoft, Tulsa, OK, USA) software to calculate means and standard deviations. To assess the significance of differences between the means, Tukey’s test was applied, with a significance level of *p* ≤ 0.05. A three-way analysis of variance (ANOVA) was employed to examine the combined impact of bacterial strains, dose, and type of protein isolate on fermented milk properties.

## 3. Results and Discussion

### 3.1. Acidity of Fermented Milk

The pH value of the control milk fermented by *L. casei* on the first day of storage was 4.08 (Table 1). The addition of 1.5% SPI and WPI did not significantly affect the active acidity of the fermented milk compared to the control sample. However, a notable increase in pH was observed in fermented milk samples LCS3 and LCW3 and samples with the addition of 1.5% and 3.0% PPI. The highest pH values were found in milk fermented with PPI, by 0.15 (LCP1.5) and 0.22 (LCP3) compared to the control group. On the 21st day of refrigerated storage, the pH values of the fermented milk remained relatively stable, ranging from 3.74 in the LCS1.5 sample to 4.03 in the LCP3 sample, showing no significant variation. The study demonstrated that storage time had a significant impact on the pH of milk fermented by *L. casei*. By day 21, the pH of the fermented milk had decreased in all experimental groups by up to 0.34 units (LCS1.5) compared to the initial measurements on day 1. 

Pelaes Vital et al. [28] investigated the effect of addition of SPI and WPI at a concentration of 1.8% on yogurt pH levels. Their findings showed that yogurts with added SPI exhibited a lower pH of 4.55 on the first day of refrigerated storage. By day 21, the control yogurts showed the most significant pH reduction, reaching 4.38, while the experimental groups with WPI and SPI maintained a pH of 4.40. The authors [28] suggested that the shorter incubation time and slightly lower pH observed in yogurt with added SPI might be associated with the prebiotic effect of soy components, such as galactooligosaccharides (e.g., raffinose). Our findings support similar conclusions for milk with SPI fermented by *L. johnsonii*, where the pH values were lower than the control group at both time points. In contrast, the experimental groups with 3% of SPI and *L. casei* did not exhibit this interaction. It can be inferred that the active acidity levels were influenced not only by the addition of SPI but also by the specific probiotic strain used for fermentation. Mashayekh et al. [29] studied the effect of soy whey-derived peptide addition (6.5, 13.0, and 17.0 mg mL^−1^) on the quality characteristics of functional yogurt. In their study, the addition of bioactive peptides significantly increased the pH of the yogurts in the experimental groups during storage compared to the control yogurt. This phenomenon can be explained by the reduction in microbial growth rate, leading to the production of fewer acidic compounds by the yogurt starter cultures, resulting in a higher pH [30].

Adding 3.0% SPI was the only treatment that significantly increased the TA of fermented milk on the first day of storage compared to the control group. In contrast, the addition of PPI (1.5%, 3.0%) and 3.0% WPI significantly reduced the TA level of the fermented milk. By day 21 of storage, the lowest TA values were observed in fermented milk with PPI (LCP1.5, LCP3). Conversely, the sample with a 3.0% addition of WPI showed the highest increase in TA, with an increment of 0.73. The control sample and the fermented milk with WPI demonstrated the highest pH levels on both experimental dates (Table 2). In contrast, the addition of 1.5% and 3.0% PPI and SPI significantly reduced the pH value in the fermented milk. For milk fermented with *L. johnsonii*, the storage time did not significantly affect the pH value. A slight increase in pH was observed in the control group and samples with added plant proteins. However, in the sample containing 1.5% WPI, the pH remained stable, while the addition of 3.0% WPI resulted in a slight decrease in pH.

For all plant-derived protein isolates, the TA levels in fermented milk increased on both day 1 and day 21 of storage compared to the control sample, with a further rise observed as the protein concentration increased. Conversely, a negative correlation was observed for WPI, where increasing the WPI content to 3.0% resulted in a slight decrease in TA. Moreover, fermented milk samples LJW1.5 and LJW3 exhibited significantly lower TA levels than the control group on day 1. Extending the storage period to 21 days led to a significant increase in TA levels, particularly in samples with a 1.5% addition of PPI and in other experimental groups containing SPI and WPI. The most noticeable increases in TA, by 0.25 and 0.29 g L^−1^, were observed in fermented milk with the addition of WPI.

In a study by Dabija et al. [31], the addition of 1.72% SPI and 1.38% PPI increased lactic acid levels in yogurts by 0.21 and 0.12 g 100 g^−1^. Similarly, Shori [32] found that adding soy extract to yogurt samples enhanced the degree of acidification during storage. In a study by Soleymanpuori et al. [33], the observed difference in the acidity of samples due to the addition of SPI was attributed to the stimulatory effect of soy proteins on the growth of *L. delbrueckii*, which led to increased lactic acid production. Furthermore, Karaman [34] reported that yogurts with the addition of 1% and 2% WPI exhibited higher lactic acid content (0.93% and 1.03%, respectively) on the 14th day of refrigerated storage compared to the control sample (0.85%), likely due to the higher dry matter and protein content. The author also noted a significant increase in titratable acidity on the 14th day, a common occurrence during storage due to the accumulation of lactic acid produced by bacteria. A three-way analysis of variance revealed that the pH and TA of fermented milk were significantly (*p* < 0.01) affected by the type of isolate and on the interaction between the bacterial strain and the type of isolate (Table 3).

### 3.2. Syneresis of Fermented Milk

In milk fermented by *L. casei* on the first day of storage, adding WPI at concentrations of 1.5% and 3.0% significantly reduced syneresis compared to all other samples (Table 1 and Table 2). The addition of 1.5% PPI increased syneresis by 17.78%, while the 3.0% addition resulted in a 23.82% increase relative to the control. After 21 days of storage, milk samples fermented by *L. casei* with 1.5% and 3.0% isolates exhibited significantly lower syneresis than on day 1. The only exception was the LCS1.5 sample, where syneresis was approximately 2.72% higher than on day 1. In contrast, significantly higher syneresis was observed in milk fermented by *L. johnsonii* on the first day of storage in milk with both 1.5% and 3.0% PPI addition. Compared to the control sample, a significant reduction in syneresis was found in milk with 1.5% and 3.0% added WPI. After 21 days of storage, all analyzed milk samples with protein additives showed a significant reduction in syneresis. Only in the control group was syneresis increased during storage.

An analysis of variance ANOVA (Table 3) indicated that the level of syneresis in fermented milk was significantly affected by the three factors analyzed (bacterial strain, isolate type, and isolate dose) as well as their interaction.

Syneresis is considered a defect, defined as the spontaneous separation of whey on the surface of fermented milk. This issue can be mitigated or eliminated by increasing the milk’s dry matter content to approximately 15% [35]. Other strategies include the use of stabilizers, such as starch, gelatin, and vegetable gums, or starter cultures that produce exopolysaccharides (EPS). In a study by Yousseef et al. [36], examining the fermentation of cow milk and pea milk mixtures using different starter cultures, it was found that increasing the concentration of pea protein promoted syneresis in most samples. Similarly, in a study by Dabija et al. [31], the addition of 1.38% PPI increased whey leakage, whereas an inverse effect was observed in yogurts with the addition of 1.72% SPI, where a slight reduction in syneresis (by 0.80%) was noted. Conversely, research by Znamirowska et al. [37] demonstrated that the use of WPC (whey protein concentrate) and WPI in the production of milk fermented by *Bifidobacterium animalis* ssp. *lactis* BB-12 resulted in acid gels with significantly higher levels of syneresis compared to milk fermented with skimmed milk powder. In contrast, Hashim et al. [38] found that yogurts enriched with WPI were softer and exhibited reduced syneresis compared to control yogurts. 

Proteins from different sources (soy, pea, or milk) typically exhibit distinct structural and functional properties, which can be influenced by pH in various complex ways [39]. The addition of these proteins to milk can disrupt the gel network formed by casein, leading to increased water migration and higher syneresis. However, with prolonged storage, protein interactions and microbial activities can stabilize the gel structure [40], resulting in reduced syneresis compared to the control samples without protein isolate addition.

### 3.3. Color of Fermented Milk

The color components presented in Table 1 for milk fermented by *L. casei* indicate that adding isolates generally did not affect the lightness (L*) on both the 1st and 21st days of storage. The exception was the LCS3 sample, which showed significant differences in L* values compared to the control and other isolate-containing samples at both time points. Thus, it can be concluded that SPI contributes to the graying of fermented milk. Soy protein preparations are typically brown-colored powder, which can influence the lightness of the product [41]. In the milk samples fermented by *L. casei*, the a* parameter remained negative at all time points (ranging from −0.77 for LCW 1.5 to −0.20 for LCP3), indicating a greenish hue. However, the addition of 3% SPI (LCS3) resulted in the highest yellow color (b*) on the first day of storage (b* = 11.96). In the other milk samples, the b* value was similar to the control group. After 21 days of storage, significantly higher b* values compared to the control were observed in the LCS3, LCP 1.5, and LCP3 samples.

The color parameters for milk fermented by *L. johnsonii* are presented in Table 2. On the first day of storage, the addition of protein isolates contributed to the milk’s graying, as indicated by the L* parameter compared to the control sample. The exception was the LJW3 sample, where L* = 93.07, significantly higher than the control (CLJ, L* = 91.68). After 21 days of storage, most samples exhibited higher L* values than the first day. During storage, plant pigments may undergo degradation processes, resulting in a lighter color of the fermented milk. These pigments can be broken down by enzymes present in the fermented milk or may degrade through natural chemical processes [42]. The a* parameter was also negative, similar to the samples with isolates fermented by *L. casei* at both time points. The b* parameter ranged from 8.40 (LJW3) to 13.93 (LJS3) on the first day of storage, indicating a yellow hue. After 21 days, the b* parameter values were significantly lower than the first day, except for the sample with WPI (LJW1.5). For a comprehensive description of milk color, further interpretation of the hue (h^0^) and color saturation (C*) values is required. The hue most similar to the control samples (CLC and CLJ) was observed in samples with the addition of 3% WPI on both test dates. Adding PPI to milk fermented by *L. casei* and *L. johnsonii* significantly lowered the h^0^ value. Color saturation (C*) describes the purity of fermented milk’s color. On day 1 of testing, milk with the addition of 1.5% and 3% SPI had a more intense and purer color than the other samples fermented by *L. casei* and *L. johnsonii*. After 21 days of storage, only the 3% addition of SPI and the 1.5% and 3% additions of PPI significantly affected color saturation.

An ANOVA analysis revealed that all color components of the analyzed milk were significantly influenced by the addition of isolates. Both the type and dose of isolate, as well as the interaction between these factors, showed significant effects (*p* < 0.01). 

The findings of this study align with those of other researchers. In a study by Drake et al. [41], the fortification of dairy yogurts with soy protein was found to impact their sensory, chemical, and microbiological properties. Sensory panelists reported an increase in color intensity with the addition of soy protein, which corresponded with an increase in instrumental yellowness (b*) and total color difference. Conversely, instrumental lightness (L*) and redness (a*) decreased. Similarly, Pelaes Vital et al. [28] found that the color of yogurts was significantly influenced by the properties of soy, resulting in samples with soy flour being darker, redder, and yellower. Likewise, Gomes da Costa et al. [43] observed that protein-enriched yogurt was darker compared to the control yogurt without protein enrichment. Jeske et al. [44] showed that pea protein-based yogurts exhibited a light creamy color and were noticeably less white in appearance compared to dairy-based yogurts.

### 3.4. Microbiological Analysis of Fermented Milk

An important aspect of this study is the evaluation of the impact of applied isolates on the viability and survival of probiotics in fermented milk. On the first day of storage, a significantly lower number of *L. casei* cells was found in milk enriched with SPI (LCS1.5 and LCS3) and in milk with 1.5% PPI (LCP1.5) compared to the *L. casei* population in the control milk (Table 1). There was no significant effect on the number of *L. casei* cells in milk with 3% PPI or WPI. In contrast, adding 1.5% WPI positively stimulated the growth of *L. casei* (*p* ≤ 0.05).

For the *L. johnsonii* cell count on the first day of storage, the lowest number of probiotic cells was found in milk with 1.5% WPI, unlike the results for *L. casei*. Notably, the number of *L. johnsonii* cells in milk with isolates did not differ significantly from the control group. Increasing the isolate concentration from 1.5% to 3% positively affected the growth of *L. johnsonii* only in milk with WPI, while an opposite effect was observed in milk with *L. casei*, where the probiotic population was significantly lower in LCW3 than in LCW1.5. In this study, only the increased PPI concentration significantly contributed to the growth of the *L. casei* population.

As storage time increased, the probiotic population generally decreased across all samples. After 21 days of storage, the number of bacterial cells decreased in all groups, ranging from 0.03 log cfu g^−1^ in LCW1.5 to 1.5 log cfu g^−1^ in LJP1.5. A significant reduction in the *L. johnsonii* population over time was observed only in milk samples with PPI (LJP1.5 and LJP3). The combination of storage time and SPI addition negatively affected the survival of *L. casei* (LCS1.5, LCS3). Additionally, the *L. casei* population in the control sample also underwent a significant reduction. In this study, adding WPI to milk fermented by *L. casei* and *L. johnsonii* did not significantly affect the survival of these strains after 21 days of storage.

Znamirowska et al. [37] reported that enriching milk with skimmed milk powder (SMP), WPI, and whey protein concentrate (WPC80) influenced the growth of *Bifidobacterium animalis* ssp. *lactis* BB-12 in fermented milk samples with increased protein content. These authors observed the highest cell counts in milk with WPC80, while significantly lower bacterial cell counts were found in milk with SMP and WPI. The fermented milk met the therapeutic minimum criterion, with bacterial cell counts exceeding 6 log cfu g⁻^1^. Similarly, Gustaw et al. [45] found that enriching milk with protein powders affected the bacterial cell counts in beverages made from both skimmed and full-fat milk powder. The highest number of *L. casei* cell counts was found in products containing casein glycomacropeptide and WPI. However, the authors noted that refrigerated storage led to a decrease in bacterial cell counts. After 21 days of refrigerated storage, the highest number of viable *L. casei* cells was observed in fermented beverages made with full-fat milk powder and the addition of 1% WPC65 (3.50 × 10⁸ cfu g⁻^1^).

### 3.5. Organoleptic Evaluation of Fermented Milk

Milk fermented by *L. casei* with the addition of PPI and SPI exhibited similar organoleptic properties on both the 1st and 21st days of storage (Figure 2a,b). However, adding WPI resulted in milk with slightly poorer consistency, less noticeable milky-creamy taste, and a lower perception of sour taste and fermentation odor. The use of SPI (especially in the LCS3.0 sample) led to the perception of off-taste at both time points. In contrast, milk fermented by *L. johnsonii* with the addition of PPI (LJP1.5 and LJP3) exhibited the worst consistency, the least noticeable milky-creamy taste, the lowest perception of sour taste and fermentation odor, and the highest perception of off-taste (Figure 2c). In other milk samples with isolates, the organoleptic characteristics were similar to the control milk, a trend that persisted on the 21st day of the study (Figure 2d). In the milk sample with SPI (LJS1.5 and LJS3), the perception of sour taste was the most intense on the 21st day compared to the other samples.

The results of the three-factor analysis of variance indicated that the sour taste of the fermented milk was influenced by both the type of isolate added and the bacterial strain used for fermentation (Table 3). In contrast, the milky-creamy taste and off-taste were affected by the type of isolate and the interaction between the fermentation strain and the isolate type. The odor of the milk was primarily influenced by the bacterial strain promoting fermentation.

Drake et al. [41] reached similar conclusions regarding the sour and sweet taste of milk fermented with the addition of soy protein. They found that soy flavors, aromas, and astringency increased with higher soy protein concentration. The increases in astringency and soy flavors likely contributed to the decrease in sweetness intensity. Additionally, yogurts with higher concentrations of soy protein had less non-fat dried milk added, resulting in a lower lactose concentration. Dried milk is not intensely sweet, but its reduction may have contributed to the decreased perception of sweetness. Furthermore, the authors concluded that in yogurts with added soy protein, sensory attributes such as thickness, chalkiness, soy odor, soy taste, and astringency increased with soy protein concentration, while ropiness, dairy taste and odor, and sweetness decreased. In the study by Hashim et al. [38], yogurts enriched with WPI exhibited superior structural characteristics. The authors found that adding WPI significantly improved the structure and texture (at 3% and 5% WPI) and enhanced the flavor (at 3% WPI) of yogurt samples compared to the control samples. Conversely, Abdul and Ouzon [46] observed that the control milk and the sample with added isolated soy proteins had higher sensory acceptability than those with added isolated pea proteins. However, when soy protein isolates and pea protein isolates were combined, the results were intermediate compared to using each protein individually.

## 4. Conclusions

Consumer demand for a more balanced diet rich in protein, including plant proteins, has led to increased popularity of such products. This study showed that adding whey, pea, and soy protein isolates to milk fermented by *L. casei* and *L. johnsonii* affects its physicochemical and organoleptic characteristics. The study’s findings indicate that the addition of 1.5% SPI and WPI did not significantly impact the acidity of fermented milk compared to the control samples, although pH increased with the inclusion of 1.5% and 3.0% PPI. Storage time had a notable effect on pH in *L. casei* fermented milk. Both the type of protein isolate and the probiotic strain used in fermentation were shown to influence syneresis levels, with WPI at 1.5% and 3.0% notably reducing syneresis in *L. casei* groups. Protein isolates also significantly affected color parameters, with SPI contributing most to the darkening of the milk. The type of protein isolate predominantly influenced the taste of the fermented milk, while the bacterial strain had a greater impact on the aroma. Furthermore, *L. casei* cell counts were lower in samples containing SPI and 1.5% PPI compared to controls. An increase in isolate concentration from 1.5% to 3.0% enhanced the growth of *L. johnsonii* in WPI-milk but reduced *L. casei* growth in higher WPI concentrations. Additionally, only higher concentrations of PPI significantly promoted *L. casei* growth. The survival rate of bacteria is of major importance for potential producers during milk storage with isolates. The addition of SPI significantly reduced the population of *L. casei* during storage, and the survival rate of *L. johnsonii* cells significantly decreased in the presence of PPI.

Further research should focus on the survival of probiotic bacteria in these matrices within a model digestive system. Since two doses of each isolate were used for each type of protein isolate, mixtures of different protein isolates might produce different effects. Additionally, many alternative plant protein sources still need to be investigated in this context.

## Figures and Tables

**Figure 1 foods-13-03160-f001:**
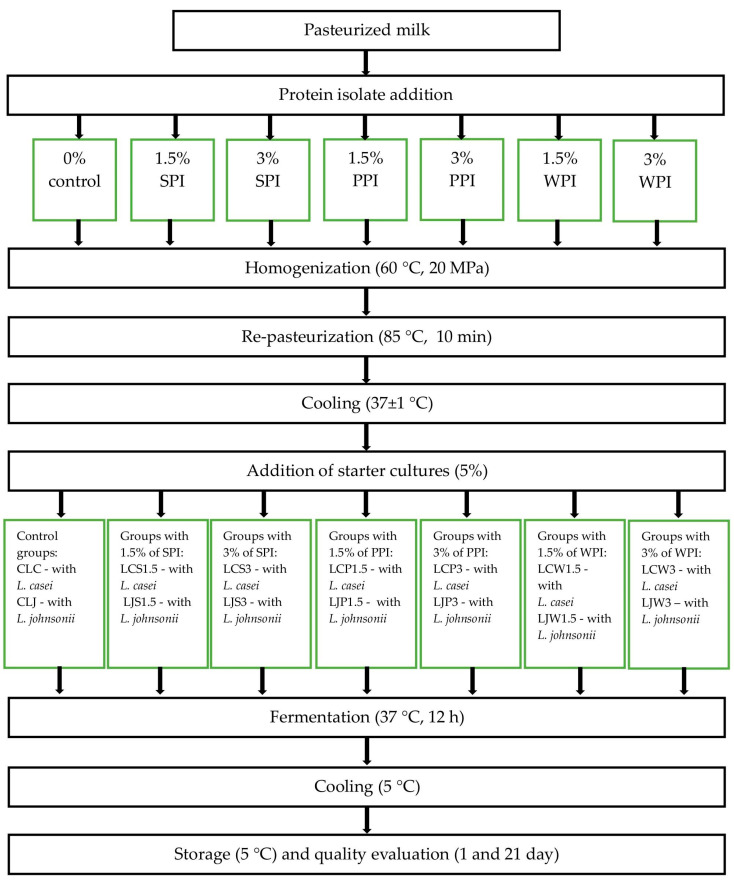
Production of fermented milk: control and with the addition of protein isolates.

**Figure 2 foods-13-03160-f002:**
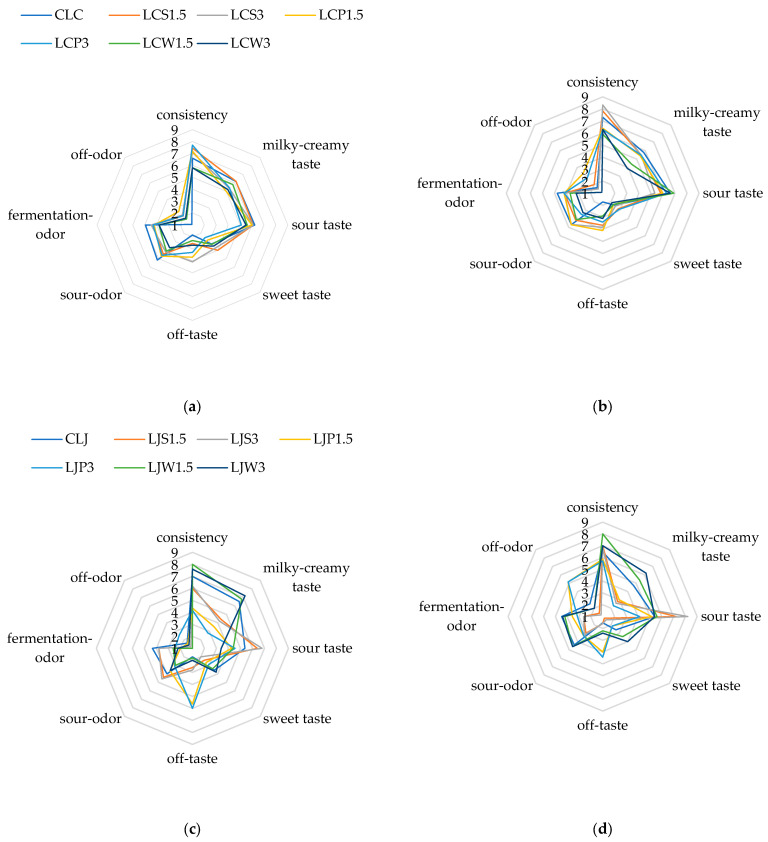
Organoleptic evaluation of milk with isolates: (**a**) fermented by *L. casei* on day 1 of cold storage; (**b**) fermented by *L. casei* on day 21 of cold storage; (**c**) fermented by *L. johnsonii* on day 1 of cold storage; (**d**) fermented by *L. johnsonii* on day 21 of cold storage. CLC—control milk fermented by *L. casei*; LCS1.5—milk with the addition of 1.5% SPI fermented by *L. casei*; LCS3—milk with the addition of 3% SPI fermented by *L. casei*; LCP1.5—milk with the addition of 1.5% PPI fermented by *L. casei*; LCP3—milk with the addition of 3% PPI fermented by *L. casei*; LCW1.5—milk with the addition of 1.5% WPI fermented by *L. casei*; LCW3—milk with the addition of 3% WPI fermented by *L. casei*; CLJ—control milk fermented by *L. johnsonii*; LJS1.5—milk with the addition of 1.5% SPI fermented by *L. johnsonii*; LJS3—milk with the addition of 3% SPI fermented by *L. johnsonii*; LJP1.5—milk with the addition of 1.5% PPI fermented by *L. johnsonii*; LJP3—milk with the addition of 3% PPI fermented by *L. johnsonii*; LJW1.5—milk with the addition of 1.5% WPI fermented by *L. johnsonii*; LJW3—milk with the addition of 3% WPI fermented by *L. johnsonii*.

**Table 1 foods-13-03160-t001:** Acidity, color, syneresis, and number of viable bacterial cells in milk fermented by *L. casei* depending on the dose and type of protein isolate and storage time.

Properties	Storage Time	Experimental Group						
		CLC	LCS1.5	LCS3	LCP1.5	LCP3	LCW1.5	LCW3
pH	1	4.08 ^aB^ ± 0.02	4.08 ^abB^ ± 0.06	4.12 ^bB^ ± 0.07	4.23 ^cB^ ± 0.04	4.30 ^dB^ ± 0.03	4.09 ^abB^ ± 0.02	4.16 ^bB^ ± 0.03
	21	3.88 ^aA^ ± 0.03	3.74 ^aA^ ± 0.39	4.01 ^aA^ ± 0.01	3.98 ^aA^ ± 0.01	4.03 ^aA^ ± 0.01	3.89 ^aA^ ± 0.01	3.92 ^aA^ ± 0.01
Total acidity, g L^−1^	1	0.90 ^cA^ ± 0.07	0.93 ^cA^ ± 0.08	1.05 ^dA^ ± 0.02	0.88 ^bA^ ± 0.05	0.84 ^bA^ ± 0.06	0.98 ^cdA^ ± 0.06	0.63 ^aA^ ± 0.02
	21	1.20 ^aB^ ± 0.11	1.43 ^cB^ ± 0.02	1.45 ^cB^ ± 0.04	1.08 ^aB^ ± 0.02	1.11 ^aB^ ± 0.01	1.35 ^bB^ ± 0.01	1.36 ^bB^ ± 0.04
Syneresis, %	1	54.03 ^dB^ ± 1.11	50.88 ^cA^ ± 0.96	48.98 ^cA^ ± 0.97	71.81 ^eB^ ± 1.50	77.85 ^fB^ ± 1.31	44.93 ^bA^ ± 1.01	40.61 ^aA^ ± 1.57
	21	50.72 ^eA^ ± 1.06	53.60 ^fB^ ± 1.66	49.41 ^dA^ ± 0.68	34.00 ^aA^ ± 2.08	34.32 ^aA^ ± 0.71	43.56 ^cA^ ± 0.23	40.60 ^bA^ ± 1.41
L*	1	93.39 ^bA^ ± 0.68	92.94 ^bA^ ± 0.79	89.02 ^aA^ ± 1.40	93.27 ^bA^ ± 0.87	92.61 ^bA^ ± 0.82	92.81 ^bA^ ± 0.72	92.84 ^bA^ ± 1.34
	21	92.72 ^bA^ ± 0.12	92.86 _bcA_ ± 0.23	89.58 ^aA^ ± 1.35	91.27 ^abA^ ± 1.23	91.59 ^bA^ ± 0.87	93.13 ^cA^ ± 0.24	93.57 ^cA^ ± 1.14
a*	1	−1.74 ^aA^ ± 0.08	−1.46 ^bA^ ± 0.12	−1.27 ^cA^ ± 0.26	−0.75 ^dA^ ± 0.09	−0.20 ^eA^ ± 0.08	−1.77 ^aA^ ± 0.09	−1.71 ^aA^ ± 0.25
	21	−1.67 ^aA^ ± 0.13	−1.44 ^bA^ ± 0.04	−1.30 ^cA^ ± 0.08	−0.82 ^dA^ ± 0.03	−0.39 ^eA^ ± 0.10	−1.67 ^aA^ ± 0.07	−1.62 ^aA^ ± 0.18
b*	1	7.69 ^aA^ ± 0.38	8.30 ^aB^ ± 0.25	11.96 ^bB^ ± 1.12	8.14 ^aA^ ± 0.36	8.44 ^aA^ ± 0.31	7.78 ^aA^ ± 0.30	7.88 ^aA^ ± 0.14
	21	7.83 ^aA^ ± 0.39	7.81 ^abA^ ± 0.03	8.64 ^bA^ ± 0.76	8.65 ^bA^ ± 0.30	8.73 ^bA^ ± 0.15	7.56 ^aA^ ± 0.08	7.59 ^aA^ ± 0.37
C*	1	7.93 ^aA^ ± 0.34	8.42 ^aB^ ± 0.23	11.40 ^bB^ ± 0.69	8.17 ^aA^ ± 0.36	8.44 ^aA^ ± 0.31	7.85 ^aA^ ± 0.48	8.13 ^aA^ ± 0.26
	21	8.03 ^bB^ ± 0.38	7.94 ^bA^ ± 0.04	8.74 ^cA^ ± 0.76	8.71 ^cA^ ± 0.29	8.75 ^cA^ ± 0.13	7.74 ^aA^ ± 0.07	7.76 ^aA^ ± 0.39
h^0^	1	102.90 ^dA^ ± 0.72	99.96 ^cA^ ± 0.97	93.44 ^aA^ ± 1.34	95.39 ^bA^ ± 0.51	91.36 ^aA^ ± 0.60	102.93 ^dA^ ± 1.12	102.54 ^dA^ ± 1.83
	21	101.96 ^eA^ ± 1.30	100.46 ^dA^ ± 0.30	98.60 ^cB^ ± 0.60	95.25 ^bA^ ± 0.33	92.63 ^aA^ ± 0.97	102.49 ^eA^ ± 0.56	102.05 ^eA^ ± 1.01
*L. casei,* log cfu g^−1^	1	11.68 ^bB^ ± 0.11	11.40 ^aB^ ± 0.19	11.34 ^aA^ ± 0.23	11.42 ^aA^ ± 0.13	11.67 ^bA^ ± 0.22	11.89 ^cA^ ± 0.18	11.64 ^bA^ ± 0.18
	21	10.87 ^abA^ ± 0.57	10.87 ^abA^ ± 0.48	10.53 ^aB^ ± 0.15	10.54 ^aA^ ± 0.56	10.60 ^aA^ ± 0.40	11.86 ^bA^ ± 0.55	11.03 ^bA^ ± 0.58

Mean ± standard deviation; 1—after 1st day of storage; 21—after 21st day of storage; ^a–f^—mean values in lines denoted by different letters differ statistically significantly (*p* ≤ 0.05) depending on the isolate type and dose; ^A–B^—mean values in columns denoted by different letters differ statistically significantly (*p* ≤ 0.05) depending on the storage time; CLC—control milk fermented by *L. casei*; LCS1.5—milk with the addition of 1.5% SPI fermented by *L. casei*; LCS3—milk with the addition of 3% SPI fermented by *L. casei*; LCP1.5—milk with the addition of 1.5% PPI fermented by *L. casei*; LCP3—milk with the addition of 3% PPI fermented by *L. casei*; LCW1.5—milk with the addition of 1.5% WPI fermented by *L. casei*; LCW3—milk with the addition of 3% WPI fermented by *L. casei*; *n* = 15 for each group.

**Table 2 foods-13-03160-t002:** Acidity, color, syneresis, and number of viable bacterial cells in milk fermented by *L. johnsonii* depending on the dose and type of protein isolate and storage time.

Properties	Storage Time	Experimental Group						
		CLJ	LJS1.5	LJS3	LJP1.5	LJP3	LJW1.5	LJW3
pH	1	4.67 ^bA^ ± 0.04	4.47 ^aA^ ± 0.03	4.41 ^aA^ ± 0.01	4.41 ^aA^ ± 0.02	4.42 ^aA^ ± 0.06	4.72 ^bA^ ± 0.03	4.74 ^bA^ ± 0.01
	21	4.68 ^cA^ ± 0.02	4.49 ^bA^ ± 0.02	4.45 ^abA^ ± 0.02	4.43 ^aA^ ± 0.02	4.45 ^aA^ ± 0.02	4.72 ^cA^ ± 0.01	4.73 ^cA^ ± 0.01
Total acidity, g L^−1^	1	0.74 ^bA^ ± 0.04	0.82 ^cA^ ± 0.06	1.01 ^dA^ ± 0.08	1.03 ^eA^ ± 0.01	1.14 ^fA^ ± 0.01	0.48 ^aA^ ± 0.01	0.42 ^aA^ ± 0.03
	21	0.75 ^aA^ ± 0.23	0.94 ^aB^ ± 0.01	1.01 ^bA^ ± 0.01	1.14 ^cB^ ± 0.02	1.25 ^dB^ ± 0.02	0.73 ^aB^ ± 0.04	0.71 ^aB^ ± 0.04
Syneresis, %	1	55.84 ^bA^ ± 1.41	57.04 ^bB^ ± 0.51	58.80 ^bB^ ± 1.50	64.15 ^cB^ ± 0.62	63.49 ^cB^ ± 1.78	52.19 ^aB^ ± 0.89	53.36 ^aB^ ± 2.16
	21	66.62 ^eB^ ± 1.25	41.20 ^bA^ ± 1.24	41.53 ^bA^ ± 2.73	61.90 ^dA^ ± 0.87	57.94 ^cA^ ± 2.58	37.52 ^aA^ ± 1.51	34.96 ^aA^ ± 2.39
L*	1	91.68 ^cA^ ± 0.48	76.59 ^aA^ ± 2.42	76.17 ^aA^ ± 2.01	75.64 ^aA^ ± 0.88	87.98 ^cA^ ± 0.96	80.25 ^bA^ ± 1.72	93.07 ^cA^ ± 1.75
	21	90.67 ^bA^ ± 1.82	90.82 ^bB^ ± 1.06	91.49 ^bB^ ± 0.38	87.84 ^aA^ ± 1.42	86.68 ^aA^ ± 0.53	91.10 ^bB^ ± 1.16	92.83 ^cA^ ± 0.24
a*	1	−1.71 ^aA^ ± 0.10	−1.26 ^cA^ ± 0.14	−1.40 ^bB^ ± 0.11	−1.04 ^cB^ ± 0.47	−0.39 ^dB^ ± 0.17	−1.32 ^cA^ ± 0.06	−1.48 ^bA^ ± 0.06
	21	−1.76 ^aA^ ± 0.11	−1.17 ^cA^ ± 0.02	−0.90 ^dA^ ± 0.05	−0.62 ^eA^ ± 0.08	−0.07 ^fA^ ± 0.06	−1.46 ^bA^ ± 0.12	−1.75 ^aB^ ± 0.02
b*	1	9.18 ^bB^ ± 0.64	11.16 ^cB^ ± 0.38	13.93 ^dB^ ± 0.45	10.47 ^cA^ ± 0.85	9.23 ^bA^ ± 0.34	8.88 ^aA^ ± 0.28	8.40 ^aA^ ± 0.48
	21	8.46 ^bA^ ± 0.55	8.35 ^bA^ ± 0.32	9.81 ^cA^ ± 0.24	9.55 ^cA^ ± 0.45	9.58 ^cA^ ± 0.17	8.42 ^bA^ ± 0.20	7.82 ^aA^ ± 0.15
C*	1	9.34 ^bA^ ± 0.64	11.24 ^dB^ ± 0.39	14.79 ^eB^ ± 1.73	10.31 ^cA^ ± 0.46	9.23 ^bA^ ± 0.34	8.98 ^aA^ ± 0.28	8.53 ^aA^ ± 0.47
	21	8.65 ^bA^ ± 0.54	8.43 ^bA^ ± 0.28	9.85 ^cA^ ± 0.24	9.57 ^cA^ ± 0.45	9.57 ^cA^ ± 0.21	8.55 ^bA^ ± 0.17	8.01 ^aA^ ± 0.15
h^0^	1	100.59 ^eA^ ± 0.67	96.43 ^cA^ ± 0.58	95.73 ^bA^ ± 0.44	94.77 ^bA^ ± 0.91	88.28 ^aB^ ± 0.13	98.47 ^dA^ ± 0.49	100.02 ^eA^ ± 1.02
	21	101.73 ^eA^ ± 0.52	97.95 ^dB^ ± 0.20	95.21 ^cA^ ± 0.22	93.72 ^bA^ ± 0.29	90.41 ^aB^ ± 0.30	101.86 ^eB^ ± 3.85	102.61 ^eB^ ± 0.20
*L. johnsonii,* log cfu g^−1^	1	10.98 ^abA^ ± 0.58	11.17 ^bA^ ± 0.68	11.10 ^abA^ ± 0.91	11.16 ^bB^ ± 0.62	11.02 ^abB^ ± 0.45	10.53 ^aA^ ± 0.22	11.21 ^bA^ ± 0.85
	21	10.47 ^aA^ ± 0.50	10.16 ^aA^ ± 0.45	9.79 ^aA^ ± 0.89	9.66 ^aA^ ± 0.86	9.65 ^aA^ ± 0.68	10.50 ^aA^ ± 0.52	10.91 ^bA^ ± 0.76

Mean ± standard deviation; 1—after 1st day of storage; 21—after 21st day of storage; ^a–f^—mean values in lines denoted by different letters differ statistically significantly (*p* ≤ 0.05) depending on the isolate type and dose; ^A–B^—mean values in columns denoted by different letters differ statistically significantly (*p* ≤ 0.05) depending on the storage time; CLJ—control milk fermented by *L. johnsonii*; LJS1.5—milk with the addition of 1.5% SPI fermented by *L. johnsonii*; LJS3—milk with the addition of 3% SPI fermented by *L. johnsonii*; LJP1.5—milk with the addition of 1.5% PPI fermented by *L. johnsonii*; LJP3—milk with the addition of 3% PPI fermented by *L. johnsonii*; LJW1.5—milk with the addition of 1.5% WPI fermented by *L. johnsonii*; LJW3—milk with the addition of 3% WPI fermented by *L. johnsonii*; *n* = 15 for each group.

**Table 3 foods-13-03160-t003:** Analysis of variance (ANOVA): *p*-values determining the effect of bacterial strains, dose, and type of protein isolate on pH, acidity, syneresis, color (L*, a*, b*, C, h^0^) consistency, milky-creamy taste, sour taste, sweet taste, off-taste, sour odor, fermentation odor, and off-odor.

Properties	Bacterial Strain	Isolate Type	Isolate Dose	Bacterial Strain ^ Isolate Type	Bacterial Strain ^ Isolate Dose	Isolate Type ^ Isolate Dose	Bacterial Strain ^ Isolate Type ^ Isolate Dose
pH	0.0110 *	0.0020 **	0.4982	0.0091 **	0.0011 **	0.1149	0.6454
Total acidity	0.0889	0.0000 **	0.8929	0.0021 **	0.5812	0.0000 **	0.0405 *
Syneresis	0.0000 **	0.0001 **	0.0304 *	0.0000 **	0.3211	0.0000 **	0.0000 **
L*	0.0000 **	0.0001 **	0.0320 *	0.0011 **	0.0113 *	0.0000 **	0.0000 **
a*	0.8487	0.0000 **	0.0000 **	0.0001 **	0.0627	0.0000 **	0.0009 **
b*	0.0000 **	0.0000 **	0.0001 **	0.1413	0.3596	0.0000 **	0.6391
C*	0.0000 **	0.0000 **	0.0000 **	0.0091 **	0.6310	0.0000 **	0.2382
h^0^	0.0000 **	0.0000 **	0.0000 **	0.0348 *	0.3976	0.0000 **	0.0000 **
Consistency	0.0808	0.0238 *	0.8871	0.0000 **	0.0318 *	0.9068	0.7404
Milky-creamy taste	0.1310	0.0003 **	0.3303	0.0001 **	0.0778	0.8897	0.4034
Sour taste	0.0066 **	0.0096 **	0.6113	0.1306	0.9031	0.7417	0.4969
Sweet taste	0.6280	0.1577	0.9305	0.0772	0.3761	0.2152	0.0856
Off-taste	0.2933	0.0001 **	0.3234	0.0111 *	0.3808	0.7178	0.6292
Sour odor	0.0115 *	0.5921	0.5600	0.4512	0.7810	0.9413	0.7102
Fermentation odor	0.0002 **	0.1851	0.0078 **	0.2523	0.3437	0.7986	0.9512
Off-odor	0.0363 *	0.1996	0.5905	0.8957	0.5160	0.9694	0.6099

Bacterial strain^type of isolate = interaction; bacterial strain^dose of isolate = interaction; type of isolate^dose of isolate = interaction; bacterial strain^type of isolate^dose of isolate = interaction; *—significant effect at *p* < 0.05; **—significant effect at *p* < 0.01.

## Data Availability

The data used to support the findings of this study can be made available by the corresponding author upon request.

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
