# Peer review of "Probiotic Milk Enriched with Protein Isolates: Physicochemical, Organoleptic, and Microbiological Properties"

_foods, 2024, doi:10.3390/foods13193160_

Round 1

Reviewer 1 Report

Comments and Suggestions for Authors

This manuscript studied the effects of the addition of three protein isolates on the physicochemical, sensory and microbiological properties of yogurt fermented with two probiotics alone. Many language problems have been identified. The introduction and discussion sections need more information, and several questions on the design of the experiments need to be answered.

 (1)       Generally, after a noun's abbreviation is defined, only its abbreviation can be used in the following text, and it cannot be defined again or the full name can appear again. This problem is very common in the whole manuscript.

(2)       The introduction section did not introduce and summarize the existing data in literature on the effects of protein isolate addition on various aspects of yogurt quality, which is the core of this article. Due to the lack of this information, the article cannot explain its own innovation. In addition, the last paragraph of the introduction seems to be unfinished.

(3)       There should be no space between the number and the percent sign.

(4)       Line 78-80  The abbreviations in parentheses should be annotated here, rather than repeatedly explaining their meanings in the subsequent discussion section.

(5)       Line 83 The brackets have appeared in the wrong position.

(6)       Line 113 Why did the author choose to culture microorganisms in a vacuum desiccator instead of an incubator?

(7)       Line 145 The author has a mistaken understanding of the relationship between significance and p-value.

(8)       Line 174-175 and other places  The author repeatedly mentions the variation in lactic acid content in the discussion, but it can be seen in the methodology that what they actually measured is the total acidity, not lactic acid. Theoretically, the two are not the same thing. 

(9)       Line 276  The names of bacteria should be in italics. Please carefully check the entire text and make the necessary corrections.

(10)   Line 312-322 In this section, the author lists some research findings from others, but does not explain the purpose of listing these results, nor how they relate to the findings of this paper.

(11)   Figure 1 The lines in the figure are too thick, affecting the aesthetic appeal.

 Comments on the Quality of English Language

The language in the text must be carefully examined and improved.

Author Response

We would like to sincerely thank you for your review, all the suggestions and comments, as well as the time you have dedicated to this process.

Comments 1:
This manuscript studied the effects of the addition of three protein isolates on the physicochemical, sensory and microbiological properties of yogurt fermented with two probiotics alone. Many language problems have been identified. The introduction and discussion sections need more information, and several questions on the design of the experiments need to be answered.

Response 1: Thank you for your constructive feedback. The manuscript has been thoroughly revised for language accuracy, and additional information has been included in both the Introduction and Discussion sections to address the points raised. Additionally, 8 references have been added to the Introduction.

Comments 2:  (1) Generally, after a noun's abbreviation is defined, only its abbreviation can be used in the following text, and it cannot be defined again or the full name can appear again. This problem is very common in the whole manuscript.

Response 2: The necessary corrections have been made throughout the manuscript to ensure that, after a noun's abbreviation is defined, only the abbreviation is used in the following text, and the full name is not repeated.

Comments 3:  (2) The introduction section did not introduce and summarize the existing data in literature on the effects of protein isolate addition on various aspects of yogurt quality, which is the core of this article. Due to the lack of this information, the article cannot explain its own innovation. In addition, the last paragraph of the introduction seems to be unfinished.

Response 3: The introduction has been revised to include additional reasons for incorporating plant-based protein sources in fermented milk production. The updated introduction now provides a more comprehensive context for the use of vegetal proteins in dairy alternatives, providing a better context for the study's innovation. Additionally, the final paragraph has been completed to clearly state the objectives of the research.

Comments 4: (3) There should be no space between the number and the percent sign.

Response 4: The spacing between the number and the percent sign has been corrected throughout the manuscript.

Comments 5: (4)  Line 78-80  The abbreviations in parentheses should be annotated here, rather than repeatedly explaining their meanings in the subsequent discussion section.

Response 5: Providing explanations for the abbreviations in parentheses at this point in the text improves clarity and avoids repetition in the discussion section. The appropriate correction has been made in the manuscript.

Comments 6: (5) Line 83 The brackets have appeared in the wrong position.

Response 6: The brackets have been moved to the end of the sentence.

Comments 7: (6)  Line 113 Why did the author choose to culture microorganisms in a vacuum desiccator instead of an incubator?

Response 7: The omission of the information regarding the use of an incubator was unintentional. The manuscript has been corrected to specify that the anaerobic incubation was conducted in a vacuum desiccator placed inside an incubator.

Corrected: Inoculation was carried out using the plate-deep method with MRS agar, followed by anaerobic incubation in a vacuum desiccator placed in an incubator (Cooled Incubator ILW 115, POL-EKO Aparatura, WodzisÅ‚aw ÅšlÄ…ski, Poland) at 37°C for 72 hours, using the GENbox anaer system (Biomerieux, Warsaw, Poland) [17].

Comments 8: (7)  Line 145 The author has a mistaken understanding of the relationship between significance and p-value.

Response 8: The sentence has been revised, and the reference to p-value has been removed.

Corrected: The addition of 1.5 % SPI and WPI did not significantly alter the active acidity of the fermented milk compared to the control sample.

Comments 9: (8)  Line 174-175 and other places  The author repeatedly mentions the variation in lactic acid content in the discussion, but it can be seen in the methodology that what they actually measured is the total acidity, not lactic acid. Theoretically, the two are not the same thing.

Response 9: The text has been revised to accurately reflect that the measurements pertain to total acidity (TA) rather than specifically lactic acid content.

Comments 10: (9)  Line 276  The names of bacteria should be in italics. Please carefully check the entire text and make the necessary corrections.

Response 10: The names of bacterial strains were reviewed and corrected throughout the text.

Comments 11: (10) Line 312-322 In this section, the author lists some research findings from others, but does not explain the purpose of listing these results, nor how they relate to the findings of this paper.

Response 11: The research findings listed in this section were included because they show similar trends to those observed in our study. Such an observation was noted.

omments 12: (11) Figure 1 The lines in the figure are too thick, affecting the aesthetic appeal.

Response 12: The thickness of the lines in Figure 1 has been adjusted to improve the overall aesthetic appeal.

Reviewer 2 Report

Comments and Suggestions for Authors

In this study, the authors studied the effect of using different protein isolate sources on the physicochemical, organoleptic and microbiological properties of probiotic milk. The study offers some interesting insights for using vegetal protein sources to enrich the probiotic milk. Some aspects should be revised by the authors as follows:

The introduction part should be improved, by indicating more reasons for incorporating vegetal protein sources in the production of fermented milk, such as sustainability, shifting to plant based products etc.

As the physico chemical characteristics of the fermented milk  are dependent on the type of heat treatment applied, it is important to know the heating conditions applied of the milk before being enriched with protein isolate and repasteurized. The authors should clarify the reasons for not working with raw milk, because it is not a standard practice to use pasteurized milk prior to addition of protein sources followed then by repasteurization.

The authors must indicate, the pH of the samples at the end of 12 h of fermentation.

The conclusion part is too general and should be rewritten highlighting the main findings.

Minor changes

Line 189 – replace inverse correlation with negative correlation

Table 3 – bacterial strain, not baterial strain.

  Author Response

We would like to sincerely thank you for your review, all the suggestions and comments, as well as the time you have dedicated to this process.

Comments 1:
The introduction part should be improved, by indicating more reasons for incorporating vegetal protein sources in the production of fermented milk, such as sustainability, shifting to plant based products etc.

Response 1: Thank you for your valuable suggestion. The Introduction has been revised to include additional reasons for incorporating plant-based protein sources in fermented milk production. The updated introduction now provides a more comprehensive context for the use of vegetal proteins in dairy alternatives. Additionally, 8 references have been added to the Introduction.

Comments 2:  As the physico chemical characteristics of the fermented milk  are dependent on the type of heat treatment applied, it is important to know the heating conditions applied of the milk before being enriched with protein isolate and repasteurized. The authors should clarify the reasons for not working with raw milk, because it is not a standard practice to use pasteurized milk prior to addition of protein sources followed then by repasteurization.

Response 2: The fermented milk was intentionally produced using microfiltered and low-temperature pasteurized milk (74°C, 15 s), from the same production batch. Our department does not have the equipment required for milk microfiltration. It was crucial for us to use milk with excellent microbiological quality to avoid changes caused by fermentation carried out by the native "wild" microflora of the milk, while also ensuring consistent chemical composition to avoid introducing an additional variable. Achieving such uniformity would be significantly more challenging with raw milk. This approach allowed us to maintain control over the experimental conditions, ensuring reliable and comparable results. Additionally, prior to the start of the research cycle, studies were conducted on the thermal stability of milk with the addition of all types and doses of protein isolates. It was verified whether repasteurization at 85°C for 10 minutes would not affect the thermal stability of all samples. The experiment passed the test, so the main research was subsequently undertaken.

Comments 3: The authors must indicate, the pH of the samples at the end of 12 h of fermentation.

Response 3: Table 1 presents the pH values of the fermented milk after fermentation. These are average values calculated based on the raw data obtained for all research groups and all experimental repetitions.

Table 1. pH of fermented milk after fermentation

Bacterial strain

Experimental group

L. casei

CLC

LCS1.5

LCS3

LCP1.5

LCP3

LCW1.5

LCW3

4.11±0.03

4.10±0.05

4.15±0.02

4.25±0.02

4.33±0.06

4.11±0.03

4.19±0.02

L. johnsonii

CLJ

LJS1.5

LJS3

LJP1.5

LJP3

LJW1.5

LJW3

4.69±0.02

4.50±0.01

4.43±0.02

4.44±0.03

4.45±0.05

4.75±0.05

4.76±0.03

Comments 4:  The conclusion part is too general and should be rewritten highlighting the main findings.

Response 4: We have rewritten the conclusion to focus on the key findings of the study, emphasizing the most significant results and their implications.

Comments 5: Line 189 – replace inverse correlation with negative correlation

Response 5: The term "inverse correlation" has been replaced with "negative correlation" as requested.

Comments 6: Table 3 – bacterial strain, not baterial strain.

Response 6: The term has been corrected to "bacterial strain" in Table 3.

Round 2

Reviewer 2 Report

Comments and Suggestions for Authors

The authors have improved significantly the quality of the manuscript and it can be considered for publication in this journal.